

# The performance of cleaner wrasse, *Labroides dimidiatus,* in a reversal learning task varies across experimental paradigms

Simon Gingins[1,2,3], Fanny Marcadier[4], Sharon Wismer[1,5], Océane Krattinger[1], Fausto Quattrini[1], Redouan Bshary[1] and Sandra A. Binning[1,6]

[1] Institut de Biologie, Université de Neuchâtel, Neuchâtel, Switzerland
[2] Department of Collective Behaviour, Max Planck Institute for Ornithology, Radolfzell, Germany
[3] Department of Biology, Universität Konstanz, Konstanz, Germany
[4] Ecole Vétérinaire Nationale de Toulouse, Toulouse, France
[5] College of Science and Engineering, James Cook University, Townsville, QLD, Australia
[6] Département de sciences biologiques, Université de Montréal, Montréal, QC, Canada

Corresponding author
Simon Gingins,
simongingins@gmail.com

## ABSTRACT

Testing performance in controlled laboratory experiments is a powerful tool for understanding the extent and evolution of cognitive abilities in non-human animals. However, cognitive testing is prone to a number of potential biases, which, if unnoticed or unaccounted for, may affect the conclusions drawn. We examined whether slight modifications to the experimental procedure and apparatus used in a spatial task and reversal learning task affected performance outcomes in the bluestreak cleaner wrasse, *Labroides dimidiatus* (hereafter "cleaners"). Using two-alternative forced-choice tests, fish had to learn to associate a food reward with a side (left or right) in their holding aquarium. Individuals were tested in one of four experimental treatments that differed slightly in procedure and/or physical set-up. Cleaners from all four treatment groups were equally able to solve the initial spatial task. However, groups differed in their ability to solve the reversal learning task: no individuals solved the reversal task when tested in small tanks with a transparent partition separating the two options, whereas over 50% of individuals solved the task when performed in a larger tank, or with an opaque partition. These results clearly show that seemingly insignificant details to the experimental set-up matter when testing performance in a spatial task and might significantly influence the outcome of experiments. These results echo previous calls for researchers to exercise caution when designing methodologies for cognition tasks to avoid misinterpretations.

## INTRODUCTION

Cognition is broadly defined as the way organisms acquire, store, process and act upon information obtained from their environment (*Shettleworth, 2010*). Determining the extent to which species or individuals integrate and respond to this information has long been a topic of scientific interest, with the main goal of understanding the origins of

human cognitive capacities (see *Penn, Holyoak & Povinelli, 2008*; *Shettleworth, 2012*; *Burkart et al., 2014*; *Bolhuis, 2015*). To this end, many researchers have adopted a comparative approach when studying the evolution of cognition: by testing a variety of species in the same cognitive tasks, researchers aim to understand how cognitive skills are distributed across taxa, and why (*Emery & Clayton, 2004*; *MacLean et al., 2012*; *Salwiczek et al., 2012*; *Burkart et al., 2014*; *Gingins & Bshary, 2016*). Similarly, testing for performance in cognitive tasks across different sexes, ages and/or populations, allows researchers to explore the physiological, ontogenetic, and environmental mechanisms underlying within-species cognitive differences (*Brown & Braithwaite, 2005*; *Wismer et al., 2014*; *Carazo et al., 2014*; *Noble, Byrne & Whiting, 2014*; *White & Brown, 2015*). While these approaches are very useful for documenting differences and similarities across groups, studying cognition is prone to a number of potential biases. Whether in a comparative context or not, failing to notice or account for these biases may lead to inappropriate conclusions with regards to the behavioral or cognitive abilities of a group.

The way an experiment is designed and conducted can have a serious impact on the performance of the test subjects. For instance, in many species, a solitary fish will naturally join the largest of two groups of conspecifics and hence numerical abilities are often tested through shoal size discrimination (*Hager & Helfman, 1991*). Such experiments are typically conducted by placing the test subject in a central compartment of an aquarium, with two different sized conspecific groups located in separate compartments at opposite ends of the aquarium, and scoring the time spent in the vicinity of one or the other groups as a measure of choice. However, the way the experiment is conducted can affect the outcome in such tests. For instance, confining the test subject into a smaller cylinder in the middle compartment (i.e., preventing it from freely moving in the empty compartment), and controlling for potential stress pheromones released by previously tested fish, improved the performance of guppies in this numeric task (*Lucon-Xiccato et al., 2017*). Performance in spatial tasks is also likely affected by different methodologies. In spatial memory tests using a radial arms maze (i.e., where a subject should enter each arm of a maze once to get a reward and avoid revisits to the same arms), the length of arms can strongly impact the performance of rats, with longer arms yielding better results (*Brown, 1990*; *Brown & Huggins, 1993*).

Ecological relevance of the task for the test species may also affect the outcome and interpretation of cognitive tests. Indeed, pigeons typically forage in the open and are known to perform poorly compared to rats when confined in a radial arms maze experiment (*Bond, Cook & Lamb, 1981*). However, when the spatial memory of pigeons is conducted in an open field environment, performance is greatly improved (*Roberts & Van Veldhuizen, 1985*; *Spetch & Edwards, 1986*). Furthermore, pigeons perform better if the food patches are located on the ground, mimicking their natural foraging habits, rather than on elevated perches (*Spetch & Edwards, 1986*). Similarly, the numerical ability of guppies is lower when tested in experimental settings consisting of dots displayed on a vertical screen (*Gatto et al., 2017*), compared to where physical objects (i.e., discs) are placed on the aquarium floor (*Bisazza, Agrillo & Lucon-Xiccato,*

*2014*). The latter situation closely matches their natural foraging habits, and might thus represent a more ecologically relevant scenario, and easier task to solve.

Here, we asked whether variations in the experimental paradigm used in a spatial task and reversal learning affected the speed at which bluestreak cleaner wrasse *Labroides dimidiatus* (hereafter "cleaners") learned to solve these cognitive tasks. Cleaners are small coral reef fish whose ecological function is to remove ectoparasites off the surfaces of so-called "client" heterospecific fishes. The complexities of this cleaning mutualism are such that cleaners have emerged as a model system for testing strategic sophistication in vertebrates with primitive brains (*Bshary & Würth, 2001*; *Bshary & Grutter, 2006*; *Bshary, 2011*; *Pinto et al., 2011*; *Gingins et al., 2013*; *Soares et al., 2014*). Numerous studies have investigated decision making in cleaners using flat Plexiglas feeding plates attached to levers which are lowered into the experimental tanks. Laboratory experiments mimicking ecologically relevant scenarios suggest that cleaners are able to solve foraging tasks using Plexiglas plates in the laboratory similarly to how they would in nature with client fishes (*Bshary & Grutter, 2002a*, *2006*; *Pinto et al., 2011*; *Gingins et al., 2013*; *Wismer et al., 2014*). For instance, some client species will wait to get serviced (i.e., resident clients), while others will swim away if not immediately tended to by the cleaner (i.e., visitor clients). This situation can be mimicked in the lab by altering the behavior of two rewarding plates: one that is removed from the tank if not inspected first (representing visitor clients) and one which remains accessible regardless of the order it is approached (representing resident clients). Cleaners quickly learn that they should give priority to the ephemeral "visitor" plate (*Bshary & Grutter, 2002a*).

Cognitive tasks that are less based on the specific ecology of cleaners, such as spatial tasks, have also used feeding plates (*Gingins & Bshary, 2016*). In these spatial tasks, the fish must learn to associate one side of the tank (i.e., left or right) with a food reward. In the current study, we compared cleaner performance in four variations of a standard spatial task and reversal learning test to evaluate the extent to which slight modifications to experimental procedure and apparatus affected cleaners' performance in solving these tasks. Two treatments involved a modification of an ecologically-relevant procedure, which might affect the relevant cue learned by the cleaner. When a cleaner chooses the correct side there are, in principle, two options regarding the behavior of the detractor plate: stay in the tank (hereafter "stay") or be removed (hereafter "lift"). In nature, cleaners give priority to visitor clients who are unwilling to wait (*Bshary & Noë, 2003*), and hence the departure of an uninspected client might be viewed as a negative outcome, prompting cleaners to attend the departed client first in the next interaction, which in the context of the spatial task, would be the wrong choice. In a lab experiment, removing the unchosen plate might interfere with learning if cleaners perceive the removal in the same way as the loss of a client foraging opportunity. If so, cleaners might be more prone to choose the side of the tank where they observed a plate leaving in the previous trial, rather than the side which offered a reward. The remaining two treatments involved modifications to the experimental apparatus, namely the opacity of the partition separating the two plates (transparent or opaque) and tank size (large versus small). Changes in partition opacity may affect cleaner performance by accentuating the
separation between the two discreet choices: the fish might perceive an opaque barrier more easily than a transparent one, and thus use this barrier as a spatial reference in the task. Finally, testing the subjects in larger tanks increases the distance needed to swim before reaching the plates. This increase in distance, and hence the time required before making a choice, also has the potential to improve the accuracy of decisions (see *Brown & Huggins, 1993*; *Chittka, Skorupski & Raine, 2009*).

## METHODS

Experiments were conducted at the Lizard Island Research Station (14°40′S 145°28′E), Australia, in August 2014 and September 2015. A total of 32 adult cleaner wrasse *L. dimidiatus,* were caught with monofilament barrier nets (10 mm stretch) and hand nets on the reefs surrounding Lizard Island. After capture, fish were brought back to the research station (within 1 h) and housed in individual aquaria with a constant flow of seawater directly pumped from the reef. Fish were each provided with a PVC tube for refuge (2 cm diameter; 10–15 cm length) and fed daily with mashed prawn smeared over the surface of Plexiglas plates (approx. 8 × 8 cm). Some individuals were used in experiments testing other cognitive abilities (i.e., biological market, feeding against preference; see *Wismer et al., 2014*; *Gingins & Bshary, 2016*; *Wismer, Grutter & Bshary, 2016*) prior to their use in our experiments. However, none of the fish were tested in a spatial task or in other tasks where they were likely to develop a side bias. Therefore, we assume that participation in previous experiments did not influence their performance in our experiments. Fish were habituated to our experimental set-up over three consecutive days before trials commenced. All experiments were carried out in accordance with the Australian Code of Practice for the Care and Use of Animals for Scientific Purposes, and under the approval of the Queensland Government (Australia) Department of Agriculture and Fisheries Animal Ethics Committee (AEC Proposal Reference Number: CA 2012/05/611). All field activities were covered by a general Queensland Fisheries Permit (2014: #82440; 2015: #149800) and GBRMPA (2014: #G11/33857.1; 2015: #G14/36625.1) permit granted to the Lizard Island Research Station.

### Spatial task and reversal learning

The experimental paradigm used in our experiments was a spatial task, whereby fish had to learn to find a food reward based on its location (left or right) in their home tank. The methods were modeled after a previous study designed to compare the performance of cleaners with closely related species (*Gingins & Bshary, 2016*). The basic experimental protocol was as follows (Fig. 1): subjects were simultaneously presented with two identical Plexiglas plates, placed next to each other approximately 10 cm apart. Between the two plates, a vertical Plexiglas partition was inserted to ensure fish could access only a single plate and allow the experimenter to determine when a definitive choice had been made. One of the two plates had an accessible food reward (mashed prawn) smeared on the back, whereas the second plate offered no food reward. At the beginning of each day of experiments, the tank was divided into a holding (approx. 1/3 of tank length) and an experimental (approx. 2/3 of tank length) compartment using

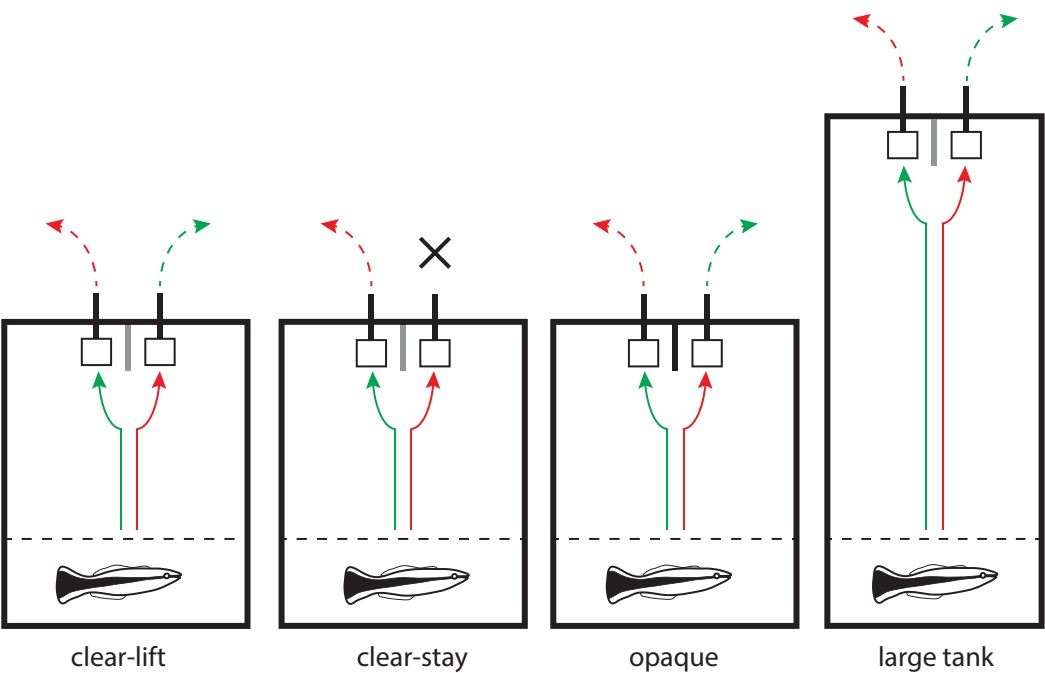

**Figure 1 The four different treatments used in the experiments.** Diagram of the four different experimental treatments used in our two-alternative forced-choice tasks. At the beginning of each trial, an opaque barrier (dotted line) separating the holding and the experimental compartments was lifted. In all treatments, the fish was presented with two identical plates separated by a small partition. One of the plates consistently had a food reward located at the back of the plate (i.e., invisible from the front). In this illustration, the correct choice (i.e., the rewarding plate) is always located on the left-hand side of the tank. The solid arrows (green = correct; red = incorrect) schematically illustrate the fish's decisions, and the dashed arrows show which plate is removed following the initial choice. In all treatments except for clear-stay, the unchosen plate was removed immediately after a choice was made. In the clear-stay treatment, the unchosen plate was only removed when the fish made an incorrect choice. The partition placed between the two plates was always transparent (grey line), except in the opaque treatment (black line). In the large tank treatment, the experiments were performed exactly as in the clear-lift treatment, but in a longer tank. Tank sizes are drawn to scale.

an opaque barrier (Fig. 1), and the fish was given 30 min to acclimate. For each trial, the barrier was lifted to allow the subject full access to the tank and the plates. A choice (left or right) was noted when the tip of the fish's snout first passed the threshold of the Plexiglas partition. The fish was given approximately 5 s to eat the food item and/or explore the experimental compartment before the barrier was placed back in the tank. The fish was kept in this holding compartment until the next trial. The trial was considered null (not taken into account) if the fish did not make a choice within 5 min. Fish were tested 20 times per day (two sessions of 10 trials) for a maximum of 100 trials (10 sessions). The task was considered solved when an individual chose the rewarding plate at least 9/10 times within a single session, 8/10 times in two consecutive sessions, or 7/10 times in three consecutive sessions. The location of the rewarding plate was randomly assigned to the right-hand side of the tank for half of the fish ($n = 16$), and to the left-hand side for the other half ($n = 16$). The location of the rewarding plate was constant throughout all the trials. All individuals that solved the initial spatial task within 10 sessions were further tested in a reversal task. Here, food plates

were changed, the location of the rewarding food plate was reversed, and trials proceeded as above. The procedures for the reversal experiment were the same as described above for the initial spatial task.

## Experimental treatments

Thirty-two cleaner wrasse were assigned to one of four variations of the spatial task described above (eight individuals per treatment). In the first treatment ("clear-lift"), the vertical partition placed between the two plates was transparent ("clear"). Thus, fish could see both plates at all times, and did not necessarily perceive the partition. As soon as the fish chose a plate, the second one was removed from the tank ("clear-lift"), which prevented the fish from accessing both plates during the trial (see Fig. 1). The fish were able to see the unselected plate being removed. Treatment two ("clear-stay") used the same clear partition as in the first treatment. However, here, the experimenter only removed the second plate when the incorrect choice (i.e., the unrewarding detractor plate) was chosen. When an individual chose the correct rewarding plate, the detractor plate remained in the tank ("stay"), and could be inspected by the cleaner. In treatment three ("opaque"), the partition separating the two food plates was made of opaque Plexiglas, which prevented the fish from seeing the second plate once a choice had been made. Here, the experimenter also removed the unchosen plate as in the "clear-lift" treatment, but the fish could not see this removal happening. These three treatments were all conducted in white plastic aquaria ($L = 37$ cm; $W = 29$ cm; $H = 30$ cm). Treatment four ("large tank") was carried out as described in the "clear-lift" treatment, but in a longer ($L = 62$ cm; $W = 26$ cm; $H = 37$ cm), glass aquarium. This setup allowed for a larger distance between the holding compartment and the plates, and thus the fish had to swim further before making a choice. Water height was maintained at approximately half of the tank height (~15cm) in all treatments.

## Statistical analysis

The number of trials to complete the task is a right-censored (i.e., maximum 100 trials), ordinal variable. Therefore, we used survival analyses to compare the number of trials needed to solve the spatial task and reversal task among the four treatments (see *Gingins & Bshary, 2016*). Our data did not meet the assumption of proportional hazards (assessed with cox.zph() from the "survival" R package), and thus non parametric log-rank tests were performed. Post-hoc pairwise comparisons were made for the reversal experiments (pairwise_survdiff() function from the "survminer" R package), and *P*-values were adjusted using the Benjamini–Hochberg method (*Benjamini & Hochberg, 1995*). All statistics were performed in R 3.3.0 (*R Core Team, 2013*). The relationship between individual's scores in the initial and reversal experiments was tested with the Kendall rank correlation coefficient. The packages "survival" (*Therneau, 2014*) and "survminer" (*Kassambra & Kosinski, 2018*) were used for the survival analyses. All data and code for the analyses are deposited in the figshare data repository (https://doi.org/10.6084/m9.figshare.5032334.v4).

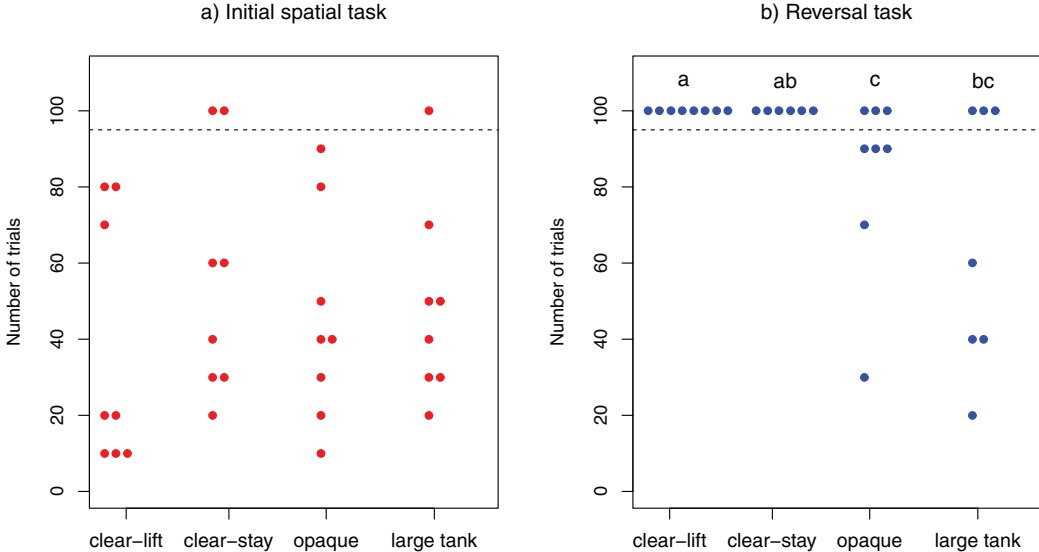

**Figure 2 Learning speed.** Number of trials required to solve the task in (A) the initial spatial discrimination task and (B) the reversal spatial discrimination task. Each dot represents one individuals tested. All individuals depicted above the dotted line failed to solve the task within the 100 allocated trials. The three individuals that did not solve the initial task were not tested in the reversal learning task. Different letters at the top indicate significant differences ($P < 0.05$) across treatments (see "Results" section for details).

## RESULTS

Overall the majority of cleaners (29/32) solved the initial spatial task, whereas only a few (9/29) managed to solve the reversal task. With the exception of three individuals (two from "clear-stay", one from "large tank"), all cleaners solved the initial spatial task within 100 trials (range : 10–90; clear-lift, mean = 37.5; clear-stay, mean = 40; opaque, mean = 45; large tank, mean = 41.43). The performance of *L. dimidiatus* in this task did not differ across treatments (log-rank test: $\chi^2_3 = 1.9$, $P = 0.595$; Fig. 2A).

In contrast, fewer fish solved the reversal test within the allocated 100 trials: 0/8 fish from the "clear-lift" and 0/6 fish from the "clear-stay" treatments solved the task, whereas 5/8 fish from the "opaque" and 4/7 fish from the "large tank treatment" solved it (range: 20–90; Opaque, mean = 74; Large Tank, mean = 40). The difference in the performance of fish in this task was significantly different among treatments (log-rank test: $\chi^2_3 = 12$, $P = 0.007$; Fig. 2B). Post-hoc pairwise comparisons revealed that cleaners performed significantly worse in the clear-lift treatment than in the opaque ($P = 0.044$) or the large tank treatment ($P = 0.044$). Similarly, in the clear-stay treatment, subjects performed significantly worse than in the opaque treatment ($P = 0.046$), and just above the threshold level when compared to the large tank treatment ($P = 0.051$).We found no significant difference between the opaque and the large tank treatments ($P = 0. 848$), or between the clear-stay and clear-lift treatments ($P = 1$, see Fig. 2B). We found no significant correlation between the speed at which cleaner's solved the initial and reversal tasks (Kendall's rank correlation: $\tau = -0.240$, $z = -1.5566$, $P = 0.1196$).

## DISCUSSION

We asked whether modifications to the experimental paradigm of typical two-alternative forced-choice tasks (spatial task and reversal learning) affected the speed of learning in the cleaner wrasse *L. dimidiatus*. We found that performance can indeed, be impaired or enhanced depending on how the experiment is implemented. Although differences in the experimental procedure and apparatus did not affect performance in the initial spatial task (Fig. 2A), the performance in reversal learning depended on the specific paradigm used (Fig. 2B). This suggests that the ability to form an initial association between food and tank location did not depend on the subtleties of the experimental design. However, the ability to form new associations in the reversal task were impeded or facilitated by specific aspects of the experimental paradigm employed. Below, we discuss potential factors which may explain our results.

### Increased distance to choice

Increasing the distance swam by the cleaners before a choice was made (i.e., large tank treatment) improved performance in the reversal learning task relative to the clear-lift and clear-stay treatments (Fig. 2B). This result parallels the findings that rats perform better in the radial arms maze with longer arms (*Brown, 1990*; *Brown & Huggins, 1993*). In this context, it appears that rats use a lax choice criterion for visiting shorter arms, thus making them more likely to revisit shorter arms (i.e., display a less accurate memory of the previously visited short arms). An increased effort to make a choice is known to improve accuracy in animals (see *Kamil & Balda, 1990*; *Zentall, Steirn & Jackson-Smith, 1990*; *Bednekoff & Balda, 1997*). A possible explanation is thus that visiting longer arms increased the investment before reinforcement, which could have lead rats to use a stricter choice criterion towards the long arms compared to the shorter ones (*Brown & Huggins, 1993*). Similarly, when the effort to reach certain arms was increased by placing barriers (rats: *Roberts & Ilersich, 1989*) or ropes (pigs: *Laughlin & Mendl, 2004*) over which animals had to climb before reaching the reward, test subjects were less likely to revisit such arms. There is a possibility that cleaners also use a lax choice criterion when distances are short. Nevertheless, in our experiments, cleaners were simultaneously presented with two plates located at an identical distance and could only "visit" one of them in each trial, hence there was no difference in investment to reach either plate, nor the possibility to revisit the same plate in the same trial. It thus seems unlikely that the choice criterion based on the investment to reach the reward played a major role here.

In our study, cleaners were conditioned to associate the removal of the barrier with the presence of a food reward. Consequently, cleaners typically darted out of the holding compartment immediately after the removal of the barrier. A longer travel distance between the holding compartment and the food plate translates into a longer computation time between the visual input of the task (i.e., plates) and the decision made. Trade-offs between speed and accuracy in both individual and collective decision making have been documented across a range of taxa (see *Franks et al., 2003*; *Chittka, Skorupski & Raine, 2009*; *Latty & Beekman, 2011*). Although performance

was similar across all treatments in the initial spatial task, it is possible that this longer computation time provided to fish in the large tank treatment facilitated the break down of a previously learned association, which is a prerequisite for success in the reversal task. Interestingly, a transparent rather than an opaque barrier between the holding and experimental compartments could also allow for a longer computation time between the visual input and the choice of the plate, yet experiments with guppies failed to find an effect for this parameter (*Gatto et al., 2017*).

## Opacity of the separating barrier

We found that the use of an opaque partition to separate the left from the right plate significantly improved the performance of cleaners in the reversal task (Fig. 2B). Interestingly, one could argue that having visual access to the unchosen plate via a transparent partition would facilitate learning, since the fish directly observes the removal of the rewarding plate following an incorrect choice, which should reinforce the negative association. However, transparent materials, such as the partition we used, are virtually non-existent in the natural world, and animals might have difficulties perceiving such a solid, yet transparent, object. In our experiments, we attributed a choice to the moment the fish's head passed on one side of the partition. If the individual was unable to perceive the transparent material, this criterion may not have been appropriate for determining a true choice by the fish: an individual may have attempted to approach the rewarding plate from the opposite side (i.e., from the side of the non-rewarding plate) and consequently run into this invisible partition, a behavior we regularly observed during the experiments. This scenario is likely given that fish generally prefer to swim close to structures such as tank walls, especially in situations which may induce anxiety (*Maximino et al., 2010*). Thus, an individual might have "known" which plate offered the food reward, but effectively "chose" the detractor plate because it approached the plates from the wrong side of the aquarium, not realizing the transparent partition would prevent it from accessing its preferred choice. This issue is intuitively more likely to occur in the reversal experiment. During the spatial task, individuals would have become conditioned to approaching the plates from the side initially offering the reward. Although the rewarding side was switched in the reversal experiment, the side preference for the approach would likely carry over from the first experiment even if fish did not receive a reward. This could explain the improved performance of cleaners in the opaque partition treatment, where the fish were able to clearly perceive the separation between the options. In other words, our criteria for what constituted a correct choice may have favored opaque barriers. If the choice criteria had been for the fish to touch the plate, it is possible that learning speed would have been similar for both opaque and transparent barriers, since many subjects did not actually touch the non-rewarding plate but went straight against the partition in the transparent treatments.

## The role of ecology

In nature, client fishes that have access to several cleaning stations are less likely to return to the same cleaner if they had been ignored during their previous visit

(*Bshary & Schäffer, 2002*). Ignoring a client can thus have negative consequences for cleaners, and we had hypothesized that cleaners might also associate the removal of the unchosen plate as the loss of a foraging opportunity (i.e., a potential client leaving the cleaning station to seek service elsewhere). Removing the unchosen plate in view of the cleaner might have thus impaired their ability to associate the positive feedback of the reward with their decision. As a result, we expected that removing the unchosen plate only when individuals made the wrong decision (i.e., the clear-stay treatment) would facilitate learning for cleaners in this task. We did not find evidence supporting this prediction, suggesting that the nature of cleaner-client interactions had little influence on performance in the spatial task. In nature, cleaners do not face noteworthy spatial challenges other than during navigation within their limited home range, and, thus do not perform better in spatial tasks than do other wrasse species (*Gingins & Bshary, 2016*). It would therefore be interesting to repeat our manipulation in an ecologically relevant task: cleaners appear to form negative associations between their decisions and the departure of a client fish in nature (*Bshary & Grutter, 2002b*), or feeding plates in the lab (*Bshary & Grutter, 2005*). One could thus repeat the study with different colors or patterns to identify the rewarding plate, rather than by its location.

## Additional sources of variation

Several additional factors might contribute to explain the results we found in this study. First, we use a relatively small sample size (eight individuals per treatment). Age, sex, rearing conditions and previous experience can affect individual performance in cognitive tasks (*Thornton & Lukas, 2012*; *Lucon-Xiccato & Bisazza, 2017*). Such inter-individual differences might impact the outcome of experiments when a limited number of subjects are tested, since small sample sizes are less likely to capture the whole spectrum of variation in performance (*Thornton & Lukas, 2012*). In cleaners, variation in performance in certain cognitive tasks has been documented across individuals from diverse developmental environments (see *Wismer et al., 2014*). Here, we controlled for this potential source of variation by testing only cleaners caught in similar habitats, but we cannot rule out that other ontogenetic factors or effects due to age might have an effect on our results. Nevertheless, we believe that the large differences we observed in the reversal experiment cannot be attributed exclusively to the small sample size used. Understanding the ecological factors driving inter-individual variation in physiological, behavioral and cognitive traits observed in this system is an interesting question in its own right which merits further investigation.

Second, while we had controlled for odour cues in a previous study using the same setup (*Gingins & Bshary, 2016*), we did not control for odour cues in the present study. Even though the odour of the food reward could potentially affect performance in a spatial task, we remain confident that odours did not influence our results for several reasons: (1) all treatments used in our comparison were performed in the same manner. An odour cue from the correct plate should help a fish solve the task, and would thus bias the results towards the null hypothesis (i.e., finding no effect of treatment). (2) If cleaners used odour cues, this spatial task should have been

straightforward for them to solve, yet in two treatments (i.e., clear-stay and clear-lift) all fish failed to solve the reversal task. The fact that we observed low performance in this task despite the potential help of odour cues suggests that odour did not have an important influence on a fish's performance. (3) We can compare the results of this experiment with that of a previous one in which odour cues had been controlled. The performance of *L. dimidiatus* in *Gingins & Bshary (2016)*, which correspond to our clear-lift treatment and where odour cues were controlled for, is similar to that of our fish. *Gingins & Bshary (2016)* had four out of eight individuals solve the task within 60 trials. Using the same criteria (60 trials), we observed five out of eight fish solve the task in our study. Therefore, it seems unlikely that odour cues helped the fish solve this spatial task.

## CONCLUSION

Our study provides additional confirmation that modifications to the way a cognitive test is designed or executed can have a significant impact on the subject's test scores (*Spetch & Edwards, 1986*; *Brown, 1990*; *Brown & Huggins, 1993*; *Gatto et al., 2017*). If different species or individuals respond differently to the experimental paradigm, this could affect interpretations derived from comparisons among groups (reviewed by *Rowe & Healy, 2014*; *Morand-Ferron, Cole & Quinn, 2016*). In our experiments, even seemingly small details such as the opacity of certain material and arena size had a significant impact on the outcome of reversal learning in our spatial task. In comparative cognition, it may be virtually impossible to design experiments in which variation in performance exactly reflects variation in cognition, particularly across species. A good understanding of the ecology of each species, and hence the cognitive challenges they naturally face, may help researchers avoid many sources of bias in individual performance. Nevertheless, biases may still go unperceived by researchers. Whether researchers are interested in comparing performance in cognitive tasks across species, populations, or individuals, it remains important to increase the number and the diversity of groups tested, to test subjects in a variety of tasks, and to reflect upon the potential influence of each species' ecology before drawing general conclusions about the cognitive abilities of a given group.

## ACKNOWLEDGEMENTS

We thank the staff at the Lizard Island Research Station for logistic support, D. Roche, A. Pinto and Z. Triki for help with fieldwork, R. A. Slobodeanu for statistical advice, and three reviewers for their useful comments on the manuscript.

### Funding

This study was supported by funding from the Swiss Science Foundation to Redouan Bshary, the Fonds de Recherche du Québec – Nature et Technologies to Sandra A. Binning and a Lizard Island Reef Research Foundation doctoral fellowship to Sharon Wismer.

The funders had no role in study design, data collection and analysis, decision to publish, or preparation of the manuscript.

### Grant Disclosures
The following grant information was disclosed by the authors:
Swiss Science Foundation.
Fonds de Recherche du Québec – Nature et Technologies.
Lizard Island Reef Research Foundation.

### Competing Interests
The authors declare that they have no competing interests.

### Author Contributions
- Simon Gingins conceived and designed the experiments, performed the experiments, analyzed the data, prepared figures and/or tables, authored or reviewed drafts of the paper, approved the final draft.
- Fanny Marcadier conceived and designed the experiments, performed the experiments, analyzed the data, authored or reviewed drafts of the paper, approved the final draft.
- Sharon Wismer performed the experiments, prepared figures and/or tables, authored or reviewed drafts of the paper, approved the final draft.
- Océane Krattinger performed the experiments, authored or reviewed drafts of the paper, approved the final draft.
- Fausto Quattrini performed the experiments, authored or reviewed drafts of the paper, approved the final draft.
- Redouan Bshary conceived and designed the experiments, authored or reviewed drafts of the paper, approved the final draft.
- Sandra A. Binning performed the experiments, analyzed the data, authored or reviewed drafts of the paper, approved the final draft.

### Animal Ethics
The following information was supplied relating to ethical approvals (i.e., approving body and any reference numbers):

All experiments were carried out in accordance with the Australian Code of Practice for the Care and Use of Animals for Scientific Purposes, and under the approval of the Queensland Government (Australia) Department of Agriculture and Fisheries Animal Ethics Committee: AEC Proposal Reference Number: CA 2012/05/611.

### Field Study Permissions
The following information was supplied relating to field study approvals (i.e., approving body and any reference numbers):

All field activities were covered by a general Queensland Fisheries Permit and GBRMPA (Great Barrier Reef Marine Park Authorities) permit granted to Lizard Island Research Station:

Queensland Fisheries Permit 2014: #82440.

GBRMPA permit 2014: #G11/33857.1.

Queensland Fisheries Permit 2015: #149800.

GBRMPA permit 2015: #G14/36625.1.

## Data Availability

Gingins, Simon; Marcadier, Fanny; Wismer, Sharon; Krattinger, Océane; Quattrini, Fausto; Bshary, Redouan; Binning, Sandra (2018): Data for: The performance of the cleaner wrasse *Labroides dimidiatus* in a reversal learning task varies across experimental setups. figshare. Fileset. https://doi.org/10.6084/m9.figshare.5032334.v4.

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
