# Peer review of "The performance of cleaner wrasse, Labroides dimidiatus, in a reversal learning task varies across experimental paradigms"

_PeerJ, doi:10.7717/peerj.4745_

## Round 0.1 · original submission · Major Revisions

Three experts in your field have now reviewed your article and each has provided clear and detailed feedback on your article and suggested ways in which you can enhance your article. I will not reiterate each of their points, but two common themes in their comments are the framing of the article and the interpretation of your results:

First, the reviewers note that your Introduction and Discussion fail to adequately recognize previous relevant studies in relation to comparative cognition research and the importance of experimental design. Furthermore, the examples you give are predominantly studies run with primates and studies of social cognition, neither of which pertain directly to your methodology. I agree with both of these points. I would think a broader overview of the literature could be provided if some of the current detail about previously-run primate studies is omitted from the Introduction.

Second, the reviewers also note some concerns with the methodology in terms of potential alternative cues (e.g., side of reward presentation and odor cues). You need to address these concerns in your Discussion and perhaps via your analyses too.

Given the soundness of your methodology, I believe that with changes to your framing and discussion to address the three reviewers' feedback, your article has the potential to be accepted for publication, although this of course is not guaranteed.

Reviewer 1 ·

Basic reporting

I have carefully read the MS entitled “The performance of cleaner wrasse, Labroides dimidiatus, in a
reversal learning task varies across experimental paradigms” by Marcadier and colleagues. The authors trained cleaner wrasses in a left-right choice followed by a reversal learning. The clear were subdivided in 4 experimental groups which differed according to details of the procedure and the apparatus. Results of the experiment showed quite clearly that reversal learning performance varies according to the experimental group.

I find the study very interesting. The effect of methodologies on cognitive abilities tests for non-human animals is possibly very important in comparative research, yet very few studies deal with the problem. Objectively, the study provides few critical data on this problem, as it is not clear what exactly caused the observed effect; however, it might be an important contribute to the journal because it will help catching the attention of other researchers on the problem.

I did not find any major issue on the experimental design, experiment execution, analysis and interpretation of the results, with one exception. It is not clear whether the authors prevented the fish to find the food reward based on odour cues. Once controlled this detail, I would be happy to see this MS published.

Writing overall is clear. I think however that the manuscript might be improved in some parts. Below, the authors will find a list of suggested improvements. Further, I think the authors ‘forget’ to cite two very important papers that investigated the very same issue on fish. I am a bit unpleased to recommend these papers as I am one of the authors. However, these suggested papers are so similar to the study here reported that I think the authors cannot avoid to report and discuss them. Details about these two papers and other suggested citations are given below.

Experimental design

Please see comments on possible odour cues.

Validity of the findings

no comments

Additional comments

L26 I am not an expert on spatial abilities but I think the terminology used by the authors might be unappropriated. The use of the term ‘spatial discrimination’ assumes that fish used a spatial strategy to solve the task, such as they learned to discriminate two positions in the tank and they learned that the reward was in a specific position; however the fish could have used non-spatial cues to solve the task; for instance, the fish could have learn to choose the stimulus at their right or left, a strategy which is based on egocentric rather than spatial cues. A correct term here might be ‘spatial task’, which assumes that the specific task is spatial, but not that the fish solved it with a spatial discrimination. I have been corrected about similar terms by researchers more expert on the topic. Thus, I think the authors should carefully check their wordings and eventually correct here and elsewhere.

L64 I think the reader would expect to see some references here.

L70-97 In this long paragraph the authors give many details about studies on primates. I see that they probably do so to make the reader understanding the importance of their works. I think they can reduce this long paragraph in few lines, and/or remove one of the examples (e.g. the one on inequity).

L99 Is there one addictional empty space between “compared” and “the”?

L98-113 Inside or after this paragraph it is the right place to discuss the previous studies on fish that manipulated small details of the procedure/apparatus and found different cognitive performance.
In the study “Gatto, E., Lucon-Xiccato, T., Savaşçı, B. B., Dadda, M., & Bisazza, A. (2017). Experimental setting affects the performance of guppies in a numerical discrimination task. Animal cognition, 20, 187-198”, my colleagues and I trained guppies in a discrimination task identical to the one of a previous study (Bisazza, A., Agrillo, C., & Lucon-Xiccato, T. (2014). Extensive training extends numerical abilities of guppies. Animal cognition, 17, 1413-1419); however, we independently changed details of the apparatus and procedure compared to the previous study. We found that most of these changes produced a reduced discrimination performance of guppies.
In the study “Lucon-Xiccato, T., Dadda, M., Gatto, E., & Bisazza, A. (2017). Development and testing of a rapid method for measuring shoal size discrimination. Animal cognition, 20, 149-157”, we modified one of the most used paradigm based on spontaneous behaviour to study discrimination in fish, the shoal choice. We found that three very small modifications in the experimental apparatus substantially improved guppies discrimination abilities. I believe that these two studies together are one of the most clear evidence that fish cognitive performance measure is affected by methodology.

L178 How can the authors be sure that none of the fish learned to find the reward following odour cues? This must be proved or the entire study might not be valid.

L190-192 I would like to see a statistical test showing that there was not effect of the initial reward side. This has been previously proved to be affect fish learning in similar spatial tasks (see this study from my lab: Miletto Petrazzini, M. E., Bisazza, A., Agrillo, C., & Lucon-Xiccato, T. (2017). Sex differences in discrimination reversal learning in the guppy. Animal Cognition).

L201 With 8 individuals per treatment, the authors might have failed to control with another issue of comparative psychology: the use of small sample size and the presence of individual differences in learning abilities (see these two papers: Thornton, A., & Lukas, D. (2012). Individual variation in cognitive performance: developmental and evolutionary perspectives. Philosophical Transactions of the Royal Society of London B: Biological Sciences, 367, 2773-2783; Lucon-Xiccato, T., & Bisazza, A. (2017). Individual differences in cognition among teleost fishes. Behavioural Processes. ).

L214 Odd parenthesis at the end of the sentence.

L237 Have the authors try to use a chi-square test to see whether there is a significant difference between the number of fish learning the task in the different conditions? This would be a stronger evidence of the effect

L248 I would add a covariance test between the score of the initial learning and the one of the reversal learning of the subjects. Maybe an ANCOVA fitted also with treatment as factor.

L275 In one of the paper that I have previously suggested (Gatto et al.), there is a result interesting for this paragraph. Fish trained with a transparent panel in front of the stimuli that forced the fish to wait in front of the stimuli before making the choice did not improve discrimination learning performance.

L304 Did you record the trials? This would allow to test the idea of fish trying to pass from one stimulus to the other despite the transparency.

L317 Spatial discrimination is not relevant for cleaners or simply less important for their ecology? Maybe the authors can use a conditional phrase.

L331 I am not sure whether “colour” properly describe the variation of the apparatus. Maybe the opacity?

Fig 1. What about drawing the transparent part of the apparatus as a empty clearer bar? This would help the reader to identify the part in the figure...

·

Basic reporting

Although this paper is well-written, I do not think the introduction and discussion show sufficient background on this topic. The effects of the details of a task on performance are not only well-studied, but could be said to be characteristic of large swathes of comparative cognition. This is not a new insight, and I am uncomfortable with the authors presenting it as such. This isn't a matter of whether the ideas and experiment in this manuscript are sufficiently "novel" (although they are presented that way by the authors), but more that it gives a false impression about the state of the field of animal cognition.

I have a couple of examples from the introduction where the authors are present commonplace ideas in comparative psychology as either unexplored or very recent (all of which can also apply to the discussion and conclusion):

64-69: The idea that species might differ in attention/motivation and this might affect performance in tests of cognition is not new, and did not originate with Lotem & Halpern (2012). Bitterman's tests of cognition across species in the 1960's were specifically controlling for this, and Alan Kamil wrote extensively about this as a possible barrier to progress to comparative studies of cognition in the 1980s (I strongly recommend his "synthetic approach to animal intelligence" from 1987, for example). This also applied to 327-330, which reads as though this insight has come about in the last couple of years.

70-73: An enormous amount of research in comparative cognition is about how slight modifications of paradigms influence performance. This is particularly true for studies of spatial memory. See for example the comparison between pigeons tested in a regular radial maze (Bond, Cook & Lamb, 1981) and in an open field maze (Spetch & Edwards 1986). Or Michael Brown's work in the early 1990's on the effect of radial maze arm length on performance in rats. These are just some early examples, there are many more from recent years. Lots of journals publish these studies frequently (e.g. The Journal of Comparative Psychology, Comparative Cognition and Behaviour Reviews, Learning & Behaviour, the Journal of the Experimental Analysis of Behavior, Journal of Experimental Psychology: Animal Learning & Cognition). These studies are interested in what these changes tell us about the mechanisms at work, so the aims might be a little different from the comparative study of cognition which you are trying to push forwards, but they are fundamentally the same.

74-97: In this vein, I think the examples used are a little odd. Not only is there a large literature dedicated to examining the effect of these kinds of manipulations, but the examples you give are also from social cognition. I understand the point you are making with this example, but I feel an example from spatial learning would be more apt.

98: The role that ecology plays (or might play) in learning is also well studied. I can see you want to refer to work on cleaner wrasse, but Kamil & Krebs and others covered similar ground in the 80s with their studies of food-storing birds (and Kamil's later work on transitive inference). Even in hardcore associative learning, it have long been known that some things can be associated together much easier than others, in ways that make ecological sense (smell+sickness or light+shock). Again, I am not objecting to this experiment, but I would like the authors to acknowledge what has come before, to prevent it feeling like they are trying to reinvent the wheel.

330-331: It might not be the authors intention, but this sounds as though good experimetnal design is a novel concept! We have known that experimental design matters in comparative cognition for the last 100 years! This shouldn't be news to anyone in animal cognition. I also think this sentence is rather unfair to your study: you can change parts of a task and still have a good experimental design. The idea that a perfectly designed experiment will unambigously get at "cognition" is just not true. Cognition is a collection of mechanisms taking in and processing multiple types of information and then guiding a wide range of behaviours. Variation in sensory systems, species typical motivations or behaviours, as well as capacity to act in terms of morphology (hard to teach a fish to point, for example), mean that no test will ever be perfect. But how animals change their performance when you change the test can tell you something about how they solved it, the mechanisms they used, and the information they learned. By pulling that apart (using experiments like this one), we can get at the mechanisms involved and then maybe explore how the properties of these mechanisms vary across species (food storing birds is a great example here). It won't be achieved by just one test (or even a battery) though.

In addition:

I do not like the term "cognitive performance". You are not directly measuring cognition, you are measuring behaviour which could be impacted by all kinds of factors (motivation, perception, morphology) of which "cognition" is only one. Just say "performance", or preferably "performance in this task".

Change "cognition test" to "spatial discrimination test", this was not a general test of cognition, but rather a specific one.

I am not sure that you can say that spatial learning is not ecologically relevant to these fish. Space is one of the few areas which is almost always ecologically relevant to (mobile) animals. Do these fish not have any locations they return to, or avoid?

Experimental design

I thought the experimental design was fine.

Validity of the findings

You seem to have found that your fish appear to learn some spatial information faster under certain conditions than under others. Rather than attributing this to differences in "cognitive performance", which suggests that cognition is homogenous and continuous, might I suggest thinking about how the fish might be solving the task and how this might be affected by the modifications you introduced (both in terms on mechanisms/processes such as discrimination or attention or associative learning, as well as the information they learn).

Space is not a single cue, and spatial tasks can be solved in a number of ways. Did the fish learn the location? Or did they learn a direction to swim? Or even the "geometry" of the arena? Do any of these impact how the fish respond to the changes? Does the poor performance in the reversal mean that they are preferentially using spatial information to solve the task (and so struggle when the spatial location of food changes)?

I think it would be helpful to look at some of the psychological work on this (as referred to in the section above) and see why, for example, people think that arm length affects radial maze performance. These studies would appear to be very relevant to the experiment you designed, and understanding more about why these differences might occur would be genuinely beneficial for moving the field forward. I seriously doubt there is such a thing as an "ideal" test of animal cognition which can be applied to any species and the results easily compared. If we are to make useful comparisons, it won't be based on performance in a test, we will need to understand the mechanisms and how they differ, and studies like this one are important for that. I just don't want to see it overlook all the progress others have already made on these topics in the last 50 years.

Additional comments

I do not think this study is flawed, but the introduction, discussion, and conclusion all ignores large areas of research on animal cognition. As a result, this manuscript describes this experiment as more novel than it actually is, and presents ideas about the influence of the task and ecology on performance, which have been discussed and widely accepted in the more psychological end of the field for decades, as fresh and new. This is unfair to the psychologists who have spent the last 50 years working hard on the kinds of experiments the authors are saying haven't really been done. It also presents an inaccurate account of the state of animal cognition in 2017, which matters as younger researchers reading this manuscript or researchers in animal behaviour unfamiliar with animal cognition will take it as fact.

I think the introduction would need to be rewritten, to take account of what has been done before, and provide an accurate overview of the field of animal cognition. As would the conclusion and parts of the discussion.

I also think reading up on what has been done before would benefit the interpretation of these results. There is a lot of work tweaking spatial tasks such as radial mazes, water-mazes, open-field mazes, as well as the details of procedures including reward sizes, reward timing, inter-trial intervals, fixed ratio versus fixed interval reinforcement, and much more. I know the authors are interested in comparing "cognition" across species, but I think looking less at performance itself and more about what performance might tell us about the mechanisms/information learned would be a good start. And also provide some much needed cross-over between experimental psychology and behavioural ecology!

P.S. Sorry to be so pedantic and editorialising a bit, but I really like animal cognition, and it pains me when people overlook all of the remarkable work done on this topic in model psychological systems like rats and pigeons over the past 50 years. This work is highly relevant for the evolutionary and ecological approach both I and the authors would like to see, but doesn't get as much attention as it should by those trying to take such an approach. Ignoring animal psychology work means that biologists interested in cognition risk slowly reinventing the wheel, and I for one would rather that didn't happen.

Reviewer 3 ·

Basic reporting

Comment 1: Clear, unambiguous, professional English is used throughout.

Comment 2: Line 28 and throughout: I have seen "two-alternative forced choice tests" written as "two-alternative forced-choice tests." May want to consider editing to stay consistent with what is in the literature.

Comment 3: Line 74-97: Although very clear and demonstrative of the authors point, I found this example to be somewhat lengthy and tangential. Additionally, both examples (Lines 74-97 and 98-113) to be somewhat primate-centric. I would recommend the authors include other examples from the literature that are shorter and more relevant. For example, see Cheney et. al. (2013) and Pintor et. al. (2014)

Comment 4: Line 99: Extra space between "compared the"


Comment 5: I would avoid the overuse of transitional phrases ("for instance" on lines 73, 93, 99).

Comment 6: Lines 123-126: I think it would be helpful to explain the specifics of the foraging tasks that can be solved, so that the reader understands how the plates are used.

Comment 7: Lines 134-135: Have these extrinsic factors been shown (empirically) to influence the outcome of tests? If so please provide citations. Also, please explain how distance swam before a decision must be made is expected to influence choice.

Comment 8: Lines 142-145: Similar to Comment 7, I think that the authors should spend more time explaining their hypotheses and predictions, so that the reader is better prepared to understand the author's interpretation of results in the discussion. In other words, please explain how "accentuating the separation between the two discreet choices and/or giving the fish a longer time" is expected to influence performance.

Comment 9: Structure, raw data, etc. seem to meet journal standards. Figure 1 is clear and well-designed.

Comment 10: Figure 2: I would consider using an asterisk to indicate statistical significance. Also, I would capitalize titles of experimental treatment remain consistent with captions, Figure 1, text, etc.

References:
Karen L. Cheney, Cait Newport, Eva C. McClure, N. Justin Marshall. (2013) "Colour vision and response bias in a coral reef fish," Journal of Experimental Biology 216: 2967-2973; doi: 10.1242/jeb.087932

Pintor, L.M., McGhee, K.E., Roche, D.P. et al. Behav Ecol Sociobiol (2014) 68: 1711. https://doi.org/10.1007/s00265-014-1779-7

Experimental design

Comment 11: This manuscript appears to be original and falls within the Aims and Scope of the journal.

Comment 12: The research question and purpose in filling a gap in knowledge is clear.

Comment 13: Investigation appears to meet high technical & ethical standards.

Comment 14: Methods are described in detail, but I would recommend providing an image or video of the plates used with the reward/bait on them (either in the MS or in supplemental information).

Comment 15: Lines 155-157: Editing required - there may be an extra “(“ ?

Comment 16: Line 155: could the authors explain what

Comment 17: Lines 155-156: Were the biological market experiments performed with actual clients or plates similar to the ones in the present experiment? Could this have influence performance at all? More of an explanation here would he helpful.

Comment 18: Line 160: Were the fish habituated to the clear partition? It seems that the clear partition could have influenced performance (e.g., lines 282-284)

Comment 19: Lines 205-206: Are the authors confident that the fish were watching/paying attention to the plate that was removed? I can imagine a sceniero where the subject made a correct choice and begin eating immediately with its attention directed away from the other plate while it was being removed. Perhaps this is not the case, but without a video or images it is unclear.

Validity of the findings

Comment 20: Results are beneficial to the literature

Comment 21: The statistical analyses used seem appropriate, however I do not have experience using these specific tests and would therefore defer to the expertise of the editor and other reviewer(s).

Comment 22: Conclusions are well stated, but I think could be better linked to the original questions/predictions (by providing more information in the introduction, see comment 7).

Comment 23: Line 282: The subjects would only form an association if they observed the other plate being removed (see comment 19).

Comment 24: Lines 267-269: This may be speculation, but I am wondering if an alternative explanation could be that the subjects could see which plate contained the reward from a further distance (i.e. when it could see both plates from further away and see which was baited)?

Comment 25: Line 287-289: Do the authors have any inclination as to whether this happened in their experiment? Did they observe subjects frequently running into the transparent partition?

Comment 26: Line 287-294: This description is slightly unclear: In Figure 1. it appears that the fish always approach the choice from the front. Do the fish instead deviate to one side of the aquarium and follow the wall, ending up in a choice because of their initial inclination to follow a wall?

Additional comments

I found this paper to be an enjoyable read and think it is a welcome contribution to the literature. I think it could be a bit stronger by including a few relevant references, images/videos, and a more detailed explanation of your predictions in the introduction (i.e., how and why would changing each of these specific parts of the experimental procedure result in differences in the performance of subjects in your task).

---

## Round 0.2 · Minor Revisions

Two of the original reviewers for your manuscript have provided feedback on your revision, and I have reviewed it thoroughly myself. I wish to commend you on the considerable edits you have clearly made, which have enhanced the clarity of the article and set it against the previous literature nicely.

In addition to Reviewer 3's comments, I have a few additional comments that I would like you to address. If you can address these it is very likely that I will be able to accept your article for publication without further review. Here are my comments:

I’m not sure the sentence (lines 54-56) “Cognitive abilities are typically assessed in terms of presence/absence of a certain cognitive process like transitive inference or imitation learning, or through the speed at which individuals learn the correct solution in a specific context.” adds much to this paragraph and could be deleted.

Line 136 – were all your fish the same size/weight?

Line 220 – the range in number of trials for the fish to solve the spatial trials i.e. 10-90 is large. Do you have any ideas about what might underlie this individual variation? (You touch on this a little in your Discussion, but it would be nice to expand this a little I think.) Also, was there any relation between the speed with which the fish solve the spatial task (i.e. number of trials) and their likelihood for success in the reversal task?

Line 224 – I think it would be helpful, before breaking down the results by treatment, to provide the overall number of fish that did not solve this task in 100 trials as you did for the spatial task (i.e. 32/32 for the spatial task and 9/32 for the reversal task).

Lines 256-258 – costs incurred in other ways (aside from distance) have been shown to improve memory too see e.g., Zentall et al. 1990 Journal of Experimental Psychology: Animal Behavior Processes 16, 358–371; Bednekoff and Balda 1997 Animal Behaviour 54, 691-698; Laughlin and Mendl 2004 Animal Behaviour 68(1), 97-103.

Reviewer 1 ·

Basic reporting

No comments

Experimental design

No comments

Validity of the findings

No comments

Additional comments

The authors modified the manuscript following my suggestions or they clearly explained their experimental choices.

I think the manuscript is focussed on the interesting point. I do not have other methodological concerns and the language is clear and easy to follow.

I think this interesting study can now be considered for pubblication.

Reviewer 3 ·

Basic reporting

- I believe that the authors have addressed my previous concerns and have improved the introduction (background/context, hypotheses, etc). I noticed just a few minor things upon review:

- Line 63: I think that the sentence "For instance, numerical abilities in fishes are often tested through shoal size discrimination as in many species, a solitary fish will naturally join the largest of two groups of conspecifics (e.g. Hager & Helfman, 1991)" may need revision.
"As in many species" stands out as odd. Perhaps add "and" before "as in many species" ? Or move "As in many species" toward front of sentence?

- Line 71: Do the authors mean "improved performance"?
"Increased performance" strikes me as meaning increased participation vs. improvement in testing/success.

Lines 129-132: I think it is good that the authors have discussed the distance needed to swim to the plates in the introduction. However, this statement still does not explain what the prediction is: I would suggest the authors consider replacing "impact" with "increase" or "improve" or "positively affect" if this is indeed their prediction (ie "This increase in distance, and hence the time required before making a choice, also has the potential to increase the accuracy of decisions...").

- I do not see the letter codes (to indicate significance) in Figure 2. Apologies if I am missing something, but I have reviewed this figure several times and I do not see the "different letters at the top" as stated in the caption...

Experimental design

No further comments on the experimental design.

Validity of the findings

No further comments on the validity of the findings.

Additional comments

I think the authors have done a nice job of addressing my comments and discussing points presented by all reviewers.

---

## Round 0.3 · accepted · Accept

Thank you very much for responding to the few outstanding suggestions that I, and the two reviewers, had. I am happy to accept your article for publication in PeerJ.

#